# The Simple Biology of Flipons and Condensates Enhances the Evolution of Complexity

**DOI:** 10.3390/molecules26164881

**Published:** 2021-08-12

**Authors:** Alan Herbert

**Affiliations:** Unit 3412, Discovery, InsideOutBio 42 8th Street, Charlestown, MA 02129, USA; alan.herbert@insideoutbio.com

**Keywords:** Z-DNA, Z-RNA, flipons, simple repeats, condensates, G4, evolution, MYC, non-coding RNA, DNA conformation, complexity, phase separation, enhancersome, nucleosome

## Abstract

The classical genetic code maps nucleotide triplets to amino acids. The associated sequence composition is complex, representing many elaborations during evolution of form and function. Other genomic elements code for the expression and processing of RNA transcripts. However, over 50% of the human genome consists of widely dispersed repetitive sequences. Among these are simple sequence repeats (SSRs), representing a class of flipons, that under physiological conditions, form alternative nucleic acid conformations such as Z-DNA, G4 quartets, I-motifs, and triplexes. Proteins that bind in a structure-specific manner enable the seeding of condensates with the potential to regulate a wide range of biological processes. SSRs also encode the low complexity peptide repeats to patch condensates together, increasing the number of combinations possible. In situations where SSRs are transcribed, SSR-specific, single-stranded binding proteins may further impact condensate formation. Jointly, flipons and patches speed evolution by enhancing the functionality of condensates. Here, the focus is on the selection of SSR flipons and peptide patches that solve for survival under a wide range of environmental contexts, generating complexity with simple parts.

## 1. Starting Simple

The theme of this article is presented in Figure 1. Here, a nucleic acid structural motif is recognized by a structure-specific protein interaction. The nucleic acid acts as a scaffold and the protein as an anchor for cellular machines. Patches on the anchor protein provide a docking site for other proteins (Figure 1, left panel). The outcome depends on the functions of the assembled proteins, any one of which may have peptide patches for the attachment of additional proteins. In the simplest case, both the nucleic acid structures and the peptide patches are encoded by simple sequence repeats (SSRs) (Figure 1, right panel). The approach to build complex biological machines is adaptative for ever-changing environments. 

The scheme exploits the properties of SSRs that enable them to encode alternative nucleic acid structures, called flipons, to specify simple peptide patches that fold in different ways and to engage sequence-specific, single-stranded binding proteins that play multiple roles in condensate biology. We will focus in this review on the biology of flipons and peptide patches: how they can initiate and how they can promote condensate formation to optimize responses. We will discuss the role of SSRs in disease and their evolutionary impact. To introduce the concepts, we will start with a description of condensates and their role in classical genetics and then move on to SSRs and flipon genetics.

## 2. Condensates

Condensates are defined as reversible protein assemblies. As the importance of condensates has become better understood, much attention has focused on the way peptide-patch-dependent assemblies work, especially those interactions involving intrinsically disordered regions (IDRs) [1,2] and low-complexity domains (LCDs) of proteins, with the former term applying to structure and the later to sequence [3].The field is advancing fast, with the view that multivalent, low-affinity interactions mediated by IDRs allow the concentration and retention of proteins once a condensate is nucleated by a high-affinity event. Recent findings also suggest that other interactions based on IDRs and LCDs help initially localize high-affinity components to their site of action.

Condensates are often discussed in relationship to phase separation, an event where proteins partition into discrete compartments. Phase separation has many advantages. It allows concentration of protein activities within a defined milieu. This localization can affect protein function through differences in free water, salt concentrations, lipid profile, effective pH, and other chemical properties specific to that environment [4]. Initially, the formation of condensates is nucleated by high-affinity interactions and then extended by weak, multivalent, reversible interactions of high avidity [5]. Multivalent interactions mediated by IDRs and LCDs are important at both stages [6,7]. IDR appear essential to localizing transcription factors to the sites where they are observed to act in cells [8], while in other examples, LCDs are essential for seeding condensate formation [9]. With aging of condensates, the plasticity and reversible nature of their formation is sometimes lost, producing insoluble aggregates similar to those observed in many diseases [10]. The more general consequences of phase separation depend on what else is happening in the cell, particularly on the availability of each component and its ability to partition at that time. 

RNAs also influence condensate formation [11,12,13,14]. RNAs can act in a non-sequence-specific manner, with interactions often involving repetitive RNAs [12,13]. For example, the nucleolar size depends on the total concentration of RNA in the cell [15]. RNA sequence-specific interactions also occur. The formation of TDP43 (TAR DNA-binding protein) and FUS RNA-binding protein aggregates depends on the specific RNAs they engage. The RNA interactions are often dynamic, with some RNA-dependent assemblies dissolving within minutes after injection of RNase A into a cell [16].

As the grooves of double-stranded RNA are too deep and narrow to allow sequence-specific recognition [17], RNAs engage sequence-specific binding proteins (SSBP) in regions such as bulges and loops, where RNA is single stranded with the bases exposed. SSBP with high affinity can seed condensate formation and use IDR-or LCD-mediated interactions to add other components [11,12,13,14]. The effects of SSBP can vary with the expression level of the RNAs they bind. At low levels of RNA, SSBP promote condensate formation, while at high levels they can dissolve them [18]. A different role for RNA SSBP has been proposed, one in which SSBP act as molecular chaperones for aggregation-prone proteins [19]. Selection over time for RNA chaperones ensures proper protein folding and guards against irreversible protein aggregation. In this scenario, mutations in proteins such as TDP43 produce disease by disrupting the RNA contacts required for the folding of soluble protein, leading to insoluble aggregates and prion formation. 

## 3. Classical Genetics

While the distinction between sequence-non-specific and sequence-specific RNA and DNA recognition in biological processes is often highlighted, it is one that in many cases is highly nuanced. For example, classical genetics is generally considered to rely on sequence-specific recognition of RNA. Yet, the pathways involved are strongly dependent on the nucleic acid structure recognized by sets of protein machinery involved (Figure 1, left panel). The best-known example is the interaction between the transfer RNA (tRNA) anticodon and the cognate messenger RNA (mRNA) codon. The structure formed by these two RNAs is accommodated into the A-site of the ribosome, but only if there is a perfect fit. The correct codon:anticodon structure enables all that follows. The proper match is essential to the accurate transfer of information from the nucleotide space to the peptide realm. The simple requirement for a perfect fit solves the decoding problem and works for every amino acid regardless of its unique chemistry. The process of mapping from the genome to the proteome then only depends on two sets of the generic cellular machinery. One assembly line process transcribes mRNA from DNA and the other translates the mRNA product into protein. There is no need, as one proposed, for a plethora of enzymes that perform a specific ligation of one amino acid to another, with specificity analogous to that observed for proteases but serving as catalysts for the reverse reaction [20]. Further examples of where structure trumps sequence include the RNA interference silencing complex (RISC) complex and CRISPR enzymes, where the generic cellular machinery engages only if the required shape is formed by the annealing of nucleotide strands. 

Repetitive, single-stranded RNAs can also adopt alternative nucleic acid conformations that are recognized by the generic protein machinery. Alu elements (AE) form one class where nucleolus formation depends on these elements [21]. AE inverted repeats can fold back on themselves and base-pair to form a double-stranded RNA that can form either left-handed Z-RNA or right-handed A-RNA, affecting the induction of interferon responses through the assembly of melanoma differentiation-associated gene 5 (MDA5, encoded by IFIH1) into filaments [22]. 

In this review, we will focus on the SSR class of repetitive DNA that have effects on a number of biological functions within the cell and disease outcomes [23,24]. We will first describe SSRs, detailing their features relevant to condensate formation, examine the alternative conformations they form, and then explore how the complexity they enable speeds evolution.

## 4. Simple Sequence Repeats 

SSRs are repeats of sequences one to six base pairs long, comprising about 3% of the human genome [25]. They encode low-complexity peptide patches, form alternative nucleic acid structures [26], and are recognized by sequence-specific binding proteins [26] (Figure 1, right panel). SSRs differ between individuals in sequence, length, modification, and location. They are a source of evolutionary novelty [23], contributing in one study of lymphoblastoid cell lines to between 10 and 15% of the heritability of human gene expression levels attributable to common variants acting *in cis* [27].

SSRs have a higher mutation rate than IDRs, suggesting that they are favored at sites where genomic variability is advantageous to survival [28]. The repeating structure of SSRs produces variation by slipped strand mispairing during replication and through homologous recombination. One example is provided by the variation in a Z-DNA forming SSR in the Pitx1 gene enhancer that affects formation of pelvic hind fins by stickleback fish [29]. For SSRs within coding regions, mutation rates vary from 10^−7^ to 10^−3^ per generation, 1 to 10 orders of magnitude higher than the rate of point mutations, with longer repeats having the highest rate [30]. In this analysis, those SSRs scored as most variable are in the class of proteins that regulate transcription by RNA polymerase II. 

Another study surveyed the occurrence of long SSRs (SSR-L) > 175 base pairs in 1115 human genomes and a replication cohort of 2504 genomes regardless of whether they were in protein-coding regions [31]. Each genome was found to contain a median of 3 rare SSR-L and 256 common SSR-L, mostly located in non-coding regions. There was an enrichment for events that overlapped an AE (odds ratio: 1.5–2.0). An unexpected finding was that 70% of the identified SSR-L were composed of two unique nucleotides regardless of motif length. In exons, motif lengths of three and six bases are favored. In other regions, about 50% of motifs are four bases in length. Of those associates with disease, a single base mutation relative to the reference genome was observed in the SSR-L expansion. While the majority of motifs are composed of adenosine and guanosine base pairs, over 20% of SSR-L in promoter and exon regions are composed of G and C pairs, in contrast to less than 1% for all other categorical groups. The enrichment of G and C base pairs corresponds to the enrichment of Z-DNA [32] and G4-forming sequences [33] in promoter and non-coding regions. The propensity to form these alternative structures also increases as the SSR length increases [32,34], suggesting that SSR expansion is related to the selection of flipons that affect gene expression by switching conformation. 

The variations in SSR also affect the peptide patches they encode. These patches only partially overlap those of IDR. One example is provided in a study of single-amino-acid (SAA) repeats in proteins. Around 32% were encoded by SSR, most often using codons for a particular amino acid at a different frequency than used in the rest of the proteome. In contrast to SSR, other SAA repeats are encoded by a mix of codons [35]. An analysis of all SAA repeats revealed that glycine, serine, glutamic acid, proline, alanine, and glutamine SAA repeats were highly abundant, while those for cysteine, phenylalanine, valine, methionine, tyrosine, tryptophan, isoleucine, and asparagine were lower than expected, given the amino acid frequency of each in the proteome.

The SAA repeats were mixed in terms of structure. Around 15% of SAA repeats were functionally annotated domains in the P-Fam database, while about 73% were part of disordered regions present in the D2P2.pro database. In a different survey of amino acid tandem repeats (TRs), tandem repeats were present in 68.8% of humans, with the percentage decreasing in other eukaryotes and being much lower in prokaryotes [36]. In humans, 30% of TRs encoded the same amino acid but were only found in proteins with other repeat types present. Micro-repeats 1 < length (L) ≤ 3, small repeats 4 < L ≤ 15, and domain repeats ≥ 15 represented 44%, 69%, and 16% of TRs in eukaryotes, respectively. In humans, 46% of protein TRs had ≥4 distinct TR regions, with small repeats being most common (95.0% of all predicted TRs), followed by micro-repeats (87.9%) and domain repeats (47.6%) [36]. As in the previous survey [35], TRs were enriched for glycine, glutamine, seine, and proline, amino acids associated with a propensity to form disordered structures. The shorter TRs were most frequently found at amino and carboxy termini. The distribution was similar to IDRs, with short IDRs most often located in the amino terminus. However, there was only a 2.5% overlap in the UniprotKB/Swiss-Prot database between TRs and IDRs. These findings indicate that the evolutionary history of SSR and IDR are different and that they are selected for different roles. One possibility is that the mutability of SSRs offer a selective advantage by increasing condensate diversity ton improve survivability, especially against infectious agents that are also constantly evolving.

## 5. SSR Genomic Spread 

The spread of SSRs through genomes and insertion into genes by homologous recombination and Alu-associated transposition [31] can alter the nucleic acid structures. formed around the site of their insertion [26] and can also produce new protein variants. One process creates binding sites in DNA and RNA for structure-specific proteins that can anchor condensate formation. The other process adds novel peptide patches to proteins that can lead to the assembly of novel condensates. Both processes allow the discovery of novel cellular assemblies that enhance survival. The trial-and-error process builds on protein functions already honed by evolution. Rather than changing their core functionality of proteins, the SSR insertions alter the protein surface by adding low-complexity peptide patches, especially those that are observed to occur at the end of proteins [36]. The patches enable new interactions with other proteins bearing compatible patches, creating novel combinations, some with unanticipated capabilities. The condensates formed allow repurposing of protein parts already known to work, yet in a way that is easy to regulate. Modifications, such as phosphorylation and sumoylation, to one or more of the patch residues on either protein can modulate condensate assembly, with effects depending on the number and nature of amino acids affected [37]. A well-known example is the phosphorylation of the 57 copies of the human heptad repeat in the carboxy terminus domain of RNA polymerase II. Phosphorylation enables the release of the enzyme, form the promoter, and enables assembly of different condensates that are essential for the various steps of RNA transcript elongation and pre-mRNA processing [38,39].

While this strategy allows the discovery of the optimal combination of protein parts, it is not without risk. In some situations, different condensates may compete for components, impairing the functionality of one or other, or both. In other cases, futile cycles may develop where assembly and disassembly of each complex is iterative, with energy expended but without any useful work performed. In other outcomes, the assemblies may fail to turn over, requiring that additional options exist for their removal. Given all these potential problems, the way condensates form and disassemble becomes critical to switching state in response as circumstances change and ultimately to survival. It is in the initiation and termination steps of these events that flipons and peptide patches encoded by SSRs can play an important evolutionary role, with proteins that bind single-stranded SSRs in a sequence-specific manner setting thresholds that circumscribe these events. SSRs provide a facile means to vary condensate formation by context.

## 6. Flipon Genetics

SSRs can adopt unique conformations under physiological conditions [40]. They exemplify a class of sequences called flipons. Z-flipons form left-handed Z-DNA and Z-RNA conformations, G-flipons fold into four-stranded G4 quadruplex structures, and T-flipons hybridize three nucleic acid strands to form triplexes (Figure 2). Each flipon type is associated with a simple repeat sequence that favors its formation. The process is reversible, allowing a switch between the two conformational states [26]. The flips depend on a number of factors, including sequence composition, base modification, and local topological stress [40,41,42]. They are powered by enzymes that produce unwinding of the B-DNA helix, such as polymerases and helicases. The flips also occur in nucleic acid tangles where base pairing between different strands traps them in an alternative conformation. Here the switch to a non-canonical DNA or RNA structure relieves strain in duplex nucleic acids.

## 7. Z-Flipons

Z-RNA regulates both type I interferon responses by adenosine deaminase RNA specific (ADAR1, encoded by ADAR) [22] and the programmed cell death necroptosis pathway by Z-DNA binding protein 1 (ZBP1) [44]. In both cases, the left-handed helix is recognized by a structure specific, winged helix-turn-helix Zα protein domain without any base-specific contacts involved [45,46]. With ADAR1, Zα and another structure-specific, double-stranded RNA (dsRNA)-specific binding motif target the deaminase domain to modify adenosines in dsRNA, forming inosine that is treated by the downstream processes as the equivalent of guanosine. This process produces non-synonymous codon changes in a limited number of human substrates [47].

It is likely that other Z-binding proteins exists, some of which may also bind B-DNA. Simple peptide repeats, such as peptides with alternating lysine residues, have high specificity for Z-DNA when tested with a methylated polymer [48]. These repeats that are likely IDRs are present in a number of interesting proteins. One example is the DNA methyl transferase I (encoded by DNMT1) that contains a lysine–glycine repeat, 10 amino acids long. The repeat is unstructured in the native DNMT1 crystal (Protein Data Bank (PDB): 5GUV) but interacts with ubiquitin-specific peptidase 7 (USP7) in the co-crystal (PDB: 4YOC). USP7 negatively modulates the activity of DNMT1 [49]. It is also possible that the repeat serves to localize DNMT1 to regions that form Z-DNA to methylate them. Another example is the histone H2A.Z variant 2 (encoded by H2AZ2; NCBI: NP_036544.1) that contains three alternating lysine–alanine repeats and destabilizes nucleosomes.

Less clear cut are the roles of poly-lysine repeats in the large ATP-dependent SWI/SNF chromatin-remodeling proteins BRG1 (encoded by SMARCA4) and BRM (encoded by SMARCA2) (Figure 3). The SWI/SNF complex is able to eject nucleosomes from DNA. The stress of DNA negative supercoiling, previously relieved by winding the DNA around the histone octamer, now becomes available to stabilize Z-DNA (Figure 3). Here, the left-handed solenoidal winding around the nucleosome is converted into a left-handed DNA twist. Previous reports have revealed that Z-DNA formation induced by BRG1 is associated with activation of the colony-stimulating factor 1 (CSF1) gene [50] and also the transcription of the heme oxygenase 1 gene (encoded by HMOX1) induced by nuclear factor, erythroid 2 like 2 (NRF2, encoded by NFE2L2) in response to oxidative stress [51]. In the CSF1 promoter, Z-DNA formation occurs while DNA is bound by the nucleosome [52], consistent with a model where local induction of Z-formation in a Z-prone sequence by BRG1 initiates nucleosome eviction to create a region of open chromatin. As the flip from B-DNA to Z-DNA is cooperative, the propagation of Z-formation into adjacent segments will dislodge the entire octamer, releasing negative supercoiling that further promotes Z-formation in the region (Figure 3) and the binding of structure-specific proteins to that locus. The energy stored as Z-DNA is then available to promote the assembly of new protein complexes on the DNA. The flip to Z-DNA controls both the location and the timing of subsequent events, producing a switch from one genetic program to another. The flip back to B-DNA relieves topological stresses as the new condensate forms.

## 8. G-Flipons

G-flipons can fold in a number of different ways, with strands running either parallel or anti-parallel with loops, providing connections that vary in length and position. The G4 quartet is recognized in a number of different ways [53]. Crystal structures of DDX36 and RAP1 proteins bound to G4 DNA reveal that the G4 caps are contacted by hydrophobic residues present in an α-helix, with engagement of the phosphate backbone by basic residues [54,55]. RAP1 also binds B-DNA through the same helix that it uses to bind G4, but through a different face [55]. A number of other proteins initially characterized as single-stranded binders have subsequently been shown to bind G4 with high affinity. While their single-strand specificity was evident because they contain the RNA recognition motif (RRM), the role of peptide patches that recognizes G4 was not appreciated initially fell within IDR, lacking structure [56]. In these cases, the entropic cost of binding a disordered, single-stranded RNA is reduced by the RRM structure, while the entopic cost of docking to a peptide patch is lessened by the prepositioned backbone of the G4 motif [57]. Recognition of G-flipons through arginine–glycine motifs within IDR is common [37], with hydrophobic residues such as tyrosine and phenylalanine [56,58] showing different preferences for G4 RNA and G4 DNA [59]. Other modes of binding to G4 structures also exist. Fragile mental retardation protein recognizes the junction between B-DNA and the G4 structure, combining backbone contacts with base-specific ones [60].

The potential biological roles for G-flipons are various. Some are evidenced by both genetic and biochemical studies. There are mendelian diseases caused by defects in DNA repair and replication. The helicase variants involved show altered binding to G4 in vitro [61]. The 425 G4 interacting proteins recently identified using probes containing various constrained G4 structures are enriched for spliceosomes, RNA transport, RNA degradation, mRNA surveillance, DNA replication, and homologous recombination pathways. One G4-binding protein complex, the negative elongation factor (NELF), regulates gene expression in eukaryotic cells by promoting RNA polymerase pausing [62]. Other cell-based studies demonstrate the presence of nuclear G4 structures [63,64]. Collectively, the above studies suggest an important role for G-flipons in localizing cellular machines to regions where outcomes modify both normal function and disease risk [65].

## 9. Flipons as Scaffolds for Condensates

A number of insights derive from viewing flipons as scaffolds for condensates. Flipons provide a controlled way to initiate condensate formation, one subject to natural selection. The alternative conformation localizes required factors needed to regulate transcription, RNA processing, and epigenetic modification, while excluding nucleosomes and other B-DNA- and A-RNA-specific proteins that produce competing outcomes. The condensates formed then hold key intermediates in place until the multiple steps involved in many pre-mRNA processing events are complete. For example, double-stranded RNA editing must occur before splicing when an intron specifies the residues subject to modification [66]. Subsequently, splicing requires that the correct 5′ and 3′ junctions be brought together to avoid trans-splicing where RNA from two different genes is joined or the formation of circular RNAs where a downstream donor site is fused with an upstream acceptor site [67]. Splicing provides another example of competition where heterogenous nuclear ribonucleoproteins (hnRNP) contend by binding to the single-stranded conformation of SSRs through high-affinity, sequence-specific RRMs [68] to suppress splicing [69]. The timing of hnRNP engagement may then set a threshold for whether or not alternative splicing of a pre-mRNA exon occurs [26]. SSR insertion or expansion within a gene may represent a way to overcome this threshold to produce new isoforms. Modifications to DNA, RNA, and nucleosomes that favor the formation of alternative conformations by flipons are another way to overcome the competition with hnRNP for transcripts [26].

In other cases, condensates represent a way to compartmentalize flipons to prevent detrimental outcomes. For example, there are strong Z-DNA- and Z-RNA-forming sequences within the nucleolus where the high rate of transcription is sufficient to power the flip to the left-handed conformation. Potentially, Z-DNA formation could create detrimental outcomes by activating ZBP1-dependent necroptosis [70]. The three-layered nucleolus separates RNA transcription, RNA processing, and ribosome assembly into separate compartments [15]. The arrangements appear to prevent ZBP1 activation during ribosomal gene transcription and ribosomal RNA folding in normal cells, by having sufficiently high concentrations of proteins that keep the Z-flipon in the B-DNA conformation or by localizing topoisomerases to regions where flipons potentially can change conformation [71] or by excluding ZBP1 from the nucleolus. Another strategy is to incorporate within ribosomal DNA other classes of flipons that relax superhelical stress and prevent Z-formation. Here, it is of interest that ribosomes have G4-prone sequences [72] and nucleolar proteins such as nucleolin and nucleophosmin bind G4, as do a number of ribosomal proteins [73]. 

Condensates also provide a means to protect flipons from damage. This function is important as junctional regions are partially single stranded and prone to attack by nucleases. The exposed bases are also sensitive to damage by mutagens and reactive oxidative species, creating genomic instability in these regions [74]. When damage does occur, other types of condensates nucleated by proteins that recognize the specific lesion can effect DNA repair [75].

Condensates also provide a means to protect cells during stress and viral infection. The scaffolds nucleated by double-stranded RNAs can be prion like, such as those observed for MDA5, and induce a type I interferon response [76]. The formation of Z-RNA by Alu inverted repeats within host sequences provides a mechanism to localize the double-stranded editing enzyme ADAR1 to terminate responses against self-RNAs [22]. Stress granules represent reversible protein aggregates, where it is proposed that interactions between RNAs and RRMs in proteins with prion-like domains prevent the formation of insoluble accretions [19]. These structures also form Z-RNA that can bind Zα family members [77], suggesting that they may also regulate Z-dependent responses during infection and stress. In some cases, Z-RNA formation is sensed by ZBP1 that initiates assembly of the necrosome to initiate necroptosis, a form of programmed cell death [78]. A different condensate forms when the protein cyclic GMP–AMP synthase (cGAS) assembles on long polymers of B-DNA inappropriately present in the cytoplasm. The soluble mediator cyclic GMP–AMP (cGAMP) produced by the enzyme then activates the STING protein to induce a type I interferon response [79]. Each of these defensive responses reveal how a combination of binding modes affects condensate formation to ensure host survival.

## 10. The Search for Better Outcomes

Flipon sequences are spread widely throughout the genome. In humans, they were copied and pasted into many locations by retrotransposons with Alu elements inserting most often into genes [41]. Other repair and recombination mechanisms as well as replication errors also add to the total and affect the spread of SSRs. That the existing distribution is non-random throughout the genome, as discussed above, could just reflect how they arrived at their destination. Alternatively, it could reflect a process of selection where flipons provide a survival advantage by increasing the range of possible responses [80]. By assembling different machines at a location in the genome, they allow extraction of distinct subsets of information [41]. The variability in flipon settings just due to their sheer number likely means that no two cells have the same settings and that each individual cell processes and translates transcripts differently. A wide range of fitness landscapes become accessible, reflecting the variations in the localization of proteins and their interactions. Using simple repeats that form flipons and patch condensates together is a way to evolve faster process without the need to discover a specific protein mutation that is advantageous. In contrast, a process dependent on mutating protein sequence is challenging given the reuse of protein cores in different development pathways. What may work well in one tissue may produce dysfunction in another, as evidenced by the pleotropic nature of many mendelian diseases. While single-celled organisms mutate and divide to survive, multicellular organisms instead query their genome to discover whether tinkering with the way transcripts are processed produces a better innovation [41].

## 11. When Flipons and Codons Clash

Many simple repeats that undergo expansion produce autosomal dominant disease [81], providing insight into their biology. The adverse outcomes reflect the abilities of repeats to both form alternative structures and encode longer peptide patches that seed aggregates. For example, the hexameric FTD/ALS repeat sequence GGGGCC can be transcribed from both strands to form a G4 quartet from one [82] and an I-motif (Figure 2) from the other [83]. The RNA transcribed from both strands is translated without the need for a traditional AUG start codon, a process called repeat-associated non-AUG (RAN) translation [84]. The RNA produced encodes six different dipeptide protein products, depending on the reading frame, with the two containing arginine being the most toxic. Disruption of many fundamental processes by the alternative nucleic conformations, by the peptide repeats, and by loss of function of the protein have been proposed as causes of disease [85]. Here, the alternative flipon conformations locked in by these diseases nucleate condensates that are only formed transiently within normal cells. The condensates persist. The proteins involved may be critical to switching one cellular response to another or may be involved in a separate pathway that is disrupted by their sequestration. The outcomes vary with the functions of the flipons involved.

The question arises whether flipon sequences also encode peptide repeats that bind to the flipon that encodes them. The CGG repeat expansion that causes fragile X-related tremor/ataxia syndrome (FXTAS) does generate a polyglycine peptide (PPG) that appears to stabilize a G4 quartet formed by the RNA transcribed [86], with a PPG longer than nine residues by itself being insoluble [87]. In this case, the simple repeat codes for an alternative RNA conformation and a single amino acid repeat peptide that also forms higher-order protein structures. The interaction of the peptide with the RNA leads to disease. In cells, both PPG and G4 partition together into granules that stain with a G4-specific antibody. The granules also accumulate many other proteins, potentially interfering with the formation of alternative complexes that are essential for normal cell function. Here, codons and flipons clash. The two different schema for the encoding of genetic information, one that specifies the amino acid sequence and the other that influences nuclear acid conformation, directly target each other, producing a negative outcome. RNA from a simple repeat expansion causes disease in FXTAS by undergoing RAN translation to produce peptides that locks the flipon sequence encoding the peptide into an alternative conformation, preventing the production of more peptide, which leads to the accumulation of more RNA that then is translated into new peptide. The self-referential and self-sustaining nature of this system produces a futile cycle, leading to system failure.

## 12. When Flipons Hang out Together

Another question is, how do flipons get along with each other? Two Z-flipons in the same topological domain can compete with each other. The one that flips depends on how much energy is available to power the switch. At low energy, preferred sequences such as alternating d(CG)_8_ flip first. However, at higher energies, longer d(TG)_30_ sequences begin to flip [88]. The process is cooperative, absorbing all available energy. It forces shorter CG sequences back to B-DNA. This mechanism represents one way to assemble the Z-specific cellular machinery at a different location within a domain. Differences in distribution of topological stress within a domain also occur when flipons from various classes compete for the free energy [89]. An example is the human promoter for c-MYC, where only expression, not mutation, is sufficient to cause cancer [90]. The human promoter can form Z-DNA at four sites [91], G4 tetrads from different dG_3_ repeats [92], and triplex H-DNA, although whether they do so in vivo is likely, though not fully proven (Figure 4) [93]. Each flipon conformation has the potential to affect the status of the far upstream element (FUSE) that initiates the bursts of c-MYC RNA transcription appropriate to each cellular context [90]. The flipons do so by tuning the topological stress within the promoter region to control FUSE conformation. Only when FUSE is single-stranded do the sequence-specific FUSE-binding proteins (FBP) associate with the element. This interaction happens when the local negative supercoiling either is sufficient to melt open the FUSE duplex or enough to force FUSE to adopt a stretched DNA (S-DNA) conformation, with its bases twisted outward and available to engage FBP [94]. The opening of the FUSE helix is thought to vary with the conformational state of each flipon within the promoter, reflecting their epigenetic modifications and the condensates they form [89,95]. Here, sequence-specific FBPs, FUSE, flipons, and the epigenetic machinery coordinately regulate cellular c-MYC levels to control outcomes in a set diverse set of cell types exposed to a plethora of perturbations.

## 13. Evolutionary Selection at the Cellular Level

The distributed nature and high frequency of simple repeats in the genome open up new evolutionary strategies based on flipons and peptide patches. The combinations they enable make it certain that no two cells are the same. The probability of a flipon adopting an alternative conformation or seeding a condensate varies within each. The outcome depends on the cell’s current status, any past epigenetic modifications to flipons, and whether replication errors during their development expanded or contracted flipon SSRs in the genome [93]. Even neighboring cells differ in their state. Consequently, the response of each cell is an individual one, with clonal selection favoring cells most adaptive to their local environment, a process observed as individuals age [97]. The process is akin to that used by the immune system to optimize responses. Rather than involving recombination of specific loci, it likely relies on sequence or epigenetic changes within specific genomic regions that over time have proved most adaptive [28]. A subset of possible outcomes is amplified, as occurs in diseases such as cancer.

## 14. Missing Heritability

While these cellular outcomes reflect somatic selection, the question remains whether any of these changes are heritable. If so, they would represent one answer to the problem of missing heritability, where estimates of heritability from genotypic data, as measured by genome-wide association studies, do not match those derived by measuring the variability between twins [98]. Genetic effects arising from flipons, rather than codons, could in part account for this difference. Here, genomically encoded conformational variation is the underlying cause of phenotypic diversity.

One possible way somatic changes become heritable is by transmission of maternal and paternal RNAs via exosomes to the germline tissue. How this might work would vary by sex, given that fertilization involves a single ovum but many million sperm. In females, transmissible genomic change could be effected by retrotransposition of imported RNAs by enzymes such as polymerase Polθ [99], given that there is only one cell to target. Such a process would favor transmission of flipons that are encoded in RNA transcripts. In males, producing transmissible variation across the millions of sperm produced is more challenging. The sheer number of spermatogonial stem cells [100] and the speed with which sperm are produced would suggest that the RNA involved would need a sufficiently high concentration in exosomes to modify enough sperm to make a difference. RNAs that target epigenetic modification of repetitive elements such as SSRs are most likely to be effective. The process is possible through the piwi and microRNA machinery that is active in gametes [101]. The flipon epialleles created would vary in the propensity to switch from B-DNA to an alternative conformation, affecting the placement of pioneer nucleosomes that are transmitted through sperm [102]. These pioneer nucleosomes bind G- and C-rich regions of the genome, especially those within promoter regions, and bear specific epigenetic modifications that impact the chromatin organization early in embryonic development [103]. The epialleles transmitted to the embryo then can be transmitted to a subsequent generation if they are copied into the germinal tissue prior to segregation of primordial germ cells from the somatic tissue. Such outcomes are not fully accounted for by current measures of heritability as they do not require DNA sequence change. It may be possible to infer them from genomic regions where SSR variation is the highest and genetic studies show association of these loci with phenotype.

## 15. Future Directions

The biggest current hope is for ways to sense flipon conformation within cells in real time with techniques that relate switches in the nucleic acid structure to condensate formation. Identification of additional structure-specific binding proteins that recognize different classes of flipons will add to these studies. The availability of large-population studies with whole-genome sequences available and improved mapping of SSR location, length, and nucleotide variation will improve mapping of flipon conformations to phenotypes. The approach will also enable the tracking of epiallele transmission to embryos and help identify flipon conformation effects on early development. Together, the genomic studies will provide better estimates of the effects of flipon genetics on trait heritability.

## 16. Conclusions

The interplay between nucleic acids and proteins plays an important role in the formation of condensates. RNA levels and RNA sequence-specific interactions with proteins influence condensate formation. Alternative nucleic acid structures provide a mechanism to control the location and timing of when and where condensates seed. The process enables the switch from one genetic program to another. The threshold for these effects is set by SSBP, such as hnRNP, that bind to single-stranded SSR segments and alter folding, transcription, and translation.

The spread and sequence variation of SSRs throughout the genome over evolutionary time ultimately produces phenotypic diversity. Those SSRs that encode peptide patches favor the creation of new condensate combinations on which natural selection can act. Due to the multivalent, low-affinity nature of peptide patches, the discovery of advantageous assemblies may take a while. In contrast, those SSRs that encode flipons potentially create change over much shorter periods of time as their effects on nucleic acid conformation within genes are immediate. Any variation in flipon conformation is quickly recognized by the pre-existing, structure-specific cellular machinery, just as these switches are during times of stress or viral infection. The responses are rapid, context specific, and reversible, producing effects that map directly to outcomes. Due to their stochastic nature, flipon conformations will vary from cell to cell, favoring selection of those clones that are most adaptive in the current context. Through their effects on individual survival, these outcomes affect the probability of transmission of flipons to the next generation. Though SSRs have low information content, the simple parts they encode enhance the evolution of complexity in multicellular organisms by acting as switches to increase the number of combinations available to the next generation.

## Figures and Tables

**Figure 1 molecules-26-04881-f001:**
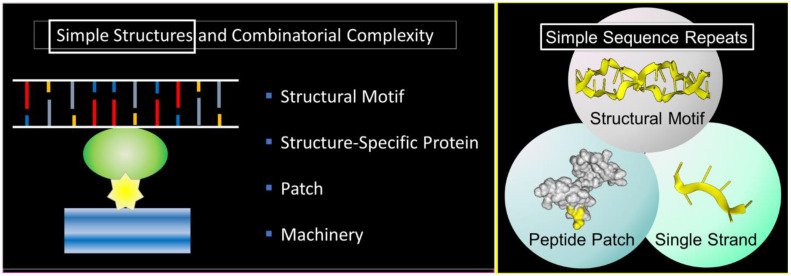
Simple nucleic acid structures and structure-specific proteins with low-complexity patches enable many combinations that enhance rapid evolution based on proven functional domains. The left panel displays flipons that form alternative nucleic acid structures under physiological conditions and are recognized by structure-specific proteins nucleate condensates that perform specific functions. The condensates assembled depend on the flipon conformation and protect the alternative structures by walling them off to maintain genome integrity. The right panel displays how simple sequence repeats impact condensate formation through structural motifs, peptide patches (highlighted in yellow), and sequence-specific interactions with single-stranded binding proteins that can also alter the condensates formed.

**Figure 2 molecules-26-04881-f002:**
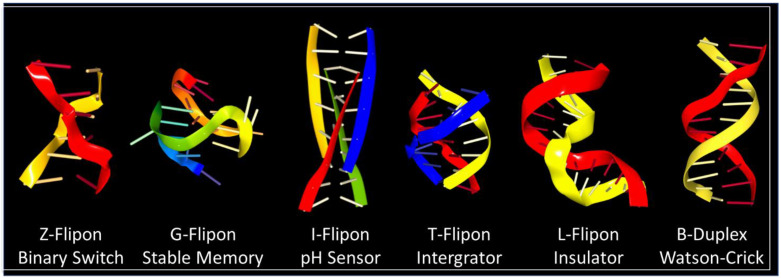
Different flipon conformations can enable different outcomes in a cell. Simple repeat sequences adopt alternative conformations, including left -handed Z-DNA and Z-RNA, G4 quadruplexes, I-motifs, and triplexes, like H-DNA formed locally by fold-back of DNA onto itself. The machines assembled on these higher-energy alternative structures differ from those made with lower-energy A-RNA, B-DNA, or on single-stranded nucleic acids. The change in flipon conformation switches outcomes. Currently, the best biologically characterized alternative conformations are Z- and G-flipons [41,43].

**Figure 3 molecules-26-04881-f003:**
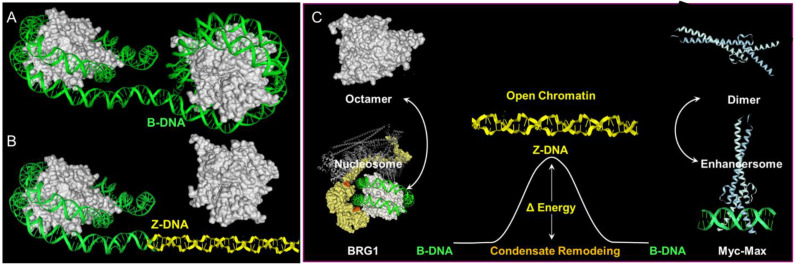
Flipons as genetic switches. The formation of alternative structures by flipons enables condensate remodeling. Here, the formation of Z-DNA is associated with nucleosome ejection and the formation of an enhancersome. The process of ejection is likely initiated by histone-remodeling complexes, such as SWI/SNF. The removal of the histone core then releases negative supercoiling that further promotes Z-DNA formation. (**A**) DNA bound by two nucleosomes. (**B**) The ejection of one nucleosome converts the left-handed writhe associated with winding of DNA around an octamer to the negative supercoiling that stabilizes left-handed Z-DNA. (**C**) In this model, the initial flip from B-DNA to Z-DNA involves lysine patches in the BRG1 protein. The segment 519–755 containing the lysine patch is not defined in the structure, while a KG patch starts at residue 1027. (Purple residues on the nucleosomes show residues 516-518 and 1020-1040 residues from PDB: 6LTJ). The Z-DNA conformation then propagates in a cooperative manner, using the energy released (ΔE) by nucleosome ejection to stabilize longer segments of DNA in the Z-DNA conformation. The Z-DNA formed then nucleates enhancersome formation and powers condensate assembly with the free energy released when DNA flips back to the B-DNA conformation. The SWI/SNF themselves are localized by transcription factors, such as NRF2 that form condensates at other locations in the enhancer region [51]. Here, the binding of the Myc-Max bZIP domain dimer is displayed.

**Figure 4 molecules-26-04881-f004:**
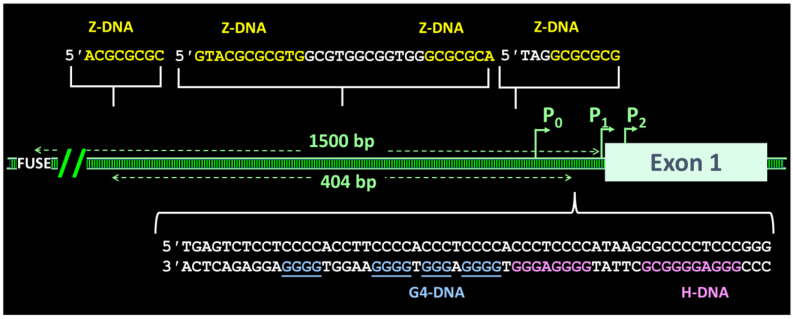
The human c-MYC promoter has many flipon sequences (based on Figure 2 of [93]). Usage of the three promoters P_0_, P_1_, and P_2_ varies by cell and tissue type. Competition exists between intramolecular H-DNA triplex formation and the sequences able to fold to form G4. These structures form on the coding strand 5′ to the promoter. A further 1.5 kilobases upstream from P1 [96] is the far upstream element (FUSE) that binds single-stranded binding proteins and regulates the release of promoter bound RNA polymerase II.

## Data Availability

The data presented in this study is from publicly available sources or is from the citations.

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
