# Peer review of "The Simple Biology of Flipons and Condensates Enhances the Evolution of Complexity"

_molecules, 2021, doi:10.3390/molecules26164881_

Round 1

Reviewer 1 Report

In this review, the author provides a timely review of alternate DNA structures (such as Z-DNA or G-quadruplexes) that can form from simple repeat sequences found within the genome, a bit about their relationship with phase separation, plus potential functions and disease associations. The review could benefit from some reorganization, with an emphasis on providing clarity to facilitate getting the reviewer’s points across.

As written, it is unclear what the review is about until Section 3. The author also appears to be using a fair amount of terminology that is not commonly used in the literature for things (e.g. alternate DNA structures) that are heavily studied (is “flipon” a term coined by the author? No one else has used this term in Pubmed abstracts), as well as a lot of non-specific pronouns (it/they), slang (e.g. “bad mash of the codons and flipons involved” p5), ambiguous references to RNA vs. DNA, etc. Thus, readers may find the review a bit confusing in areas until they figure out exactly what the author is referring to.  Once (if) that happens, some concepts introduced in the review are thought provoking.

-What are the current “big unknowns” and what does the author consider the “most important things to figure out” in the field to be? Adding this information, would strengthen the review. It is currently not clear what evidence the author discusses is correlative versus causal with respect to condensates/phase separation, and other biological phenotypes that are discussed.

-Can parallels between flipons and prions be drawn here? Is there any information from controlled systems on the rate of change between alternate states of a sequence? Are there any diseases that this has conclusively been shown to be the case?

-Abstract – Is G4 Quartests a typo?

Author Response

  1. In this review, the author provides a timely review of alternate DNA structures (such as Z-DNA or G-quadruplexes) that can form from simple repeat sequences found within the genome, a bit about their relationship with phase separation, plus potential functions and disease associations. The review could benefit from some reorganization, with an emphasis on providing clarity to facilitate getting the reviewer’s points across.

    Thanks for your comments. The review has been reorganized and have been extensively rewritten to improve clarity, with some sections expanded to provide additional information. There are now 101 references.

  1. As written, it is unclear what the review is about until Section 3.

The introduction has been rewritten to provide a better description of how the review unfolds. Also, a new panel has been added to figure 1 to provide an additional overview of the review.

  1. The author also appears to be using a fair amount of terminology that is not commonly used in the literature for things (e.g. alternate DNA structures) that are heavily studied (is “flipon” a term coined by the author? No one else has used this term in Pubmed abstracts)

    I think other authors have not included flipons in their key word list. PMID: 33154517
    is one example. See PMID: 34299306. Also, others in the field are familiar with the term and I know that it is included in other papers under review. Editors and reviewers from a number of Journals have also accepted the use of the term.

  1. as well as a lot of non-specific pronouns (it/they)slang (e.g. “bad mash of the codons and flipons involved” p5), ambiguous references to RNA vs. DNA, etc. Thus, readers may find the review a bit confusing in areas until they figure out exactly what the author is referring to. 

    Agreed – the manuscript has been rewritten to address these issues. The pronoun abuse has been corrected and the reference to “mash” removed.

  1. Once (if) that happens, some concepts introduced in the review are thought provoking.

I think you will find this version even more interesting

  1. What are the current “big unknowns” and what does the author consider the “most important things to figure out” in the field to be? Adding this information, would strengthen the review. It is currently not clear what evidence the author discusses is correlative versus causal with respect to condensates/phase separation, and other biological phenotypes that are discussed.

    I have added more examples – including many findings that have not received much attention as the framework was not in place to understand them. Hopefully that will change and there are clearly many more experiments to do! There is a big wish list under future directions.

  1. Can parallels between flipons and prions be drawn here? Is there any information from controlled systems on the rate of change between alternate states of a sequence? Are there any diseases that this has conclusively been shown to be the case?

I added examples, mostly related to the known role of condensates in interferon responses, where a clear role for Z-flipons is established, and the role of dsRNA in formation of prion like structures is known at the structural level. I also added more discussion on RNA as chaperones for protein folding that relates to the prion issue.

  1. Abstract – Is G4 Quartests a typo?

Yes! Thanks again.

Reviewer 2 Report

Dr Herbert has written a timely review about simple sequence repeats (SSRs) called flipons, their relationship to condensates and evolution. 

The review is well structured and starts by setting out how simple sequences can combine to produce complex networks of DNA structures and peptides or patches that recognize these structures. These can then help to drive functional condensates. This is followed by a consideration of the biophysical conditions that can affect formation and function of flipons and patches. Importantly, the author illustrates his arguments with good specific examples. 

His discussion of these sequences as a source of variation between cells and as fodder for  evolution is very interesting and well argued. 

In addition, the figures are simple but effective and clearly convey the points the author is making. However, I personally don’t like the black background. 

All together, I think this is a useful review of an interesting subject. The author could possibly expand his discussion of condensates to include the latest publications on the transcription and RNA processing condensates but I think he has covered his major points well.     

Author Response

Thanks for your comments. There is much more on transcription and the role of IDRs in localizing transcription factors, on SSR binding by hnRNPs (heterogeneous nuclear ribonucleoproteins) and on nucleosome remodeling including a new figure.

I have kept the dark backgrounds as the figures with such backgrounds make better slides. Also,  papers in current times  are viewed mostly electronically where a higher contrast helps with viewing molecular structures.